# Emerging Immunotherapeutic and Diagnostic Modalities in Carcinoid Tumors

**DOI:** 10.3390/molecules28052047

**Published:** 2023-02-22

**Authors:** Shahnaz D. Vellani, Anthony Nigro, Shangari Varatharajan, Lance D. Dworkin, Justin Fortune Creeden

**Affiliations:** 1Department of Medicine, University of Toledo College of Medicine and Life Sciences, Toledo, OH 43614, USA; 2Department of Cell and Cancer Biology, University of Toledo College of Medicine and Life Sciences, Toledo, OH 43614, USA; 3Department of Neurosciences, University of Toledo College of Medicine and Life Sciences, Toledo, OH 43614, USA

**Keywords:** carcinoid, immunotherapy, biomarkers, neuroendocrine, tumors

## Abstract

Evasion of innate immunity represents a frequently employed method by which tumor cells survive and thrive. Previously, the development of immunotherapeutic agents capable of overcoming this evasion has realized pronounced clinical utility across a variety of cancer types. More recently, immunological strategies have been investigated as potentially viable therapeutic and diagnostic modalities in the management of carcinoid tumors. Classic treatment options for carcinoid tumors rely upon surgical resection or non-immune pharmacology. Though surgical intervention can be curative, tumor characteristics, such as size, location, and spread, heavily limit success. Non-immune pharmacologic treatments can be similarly limited, and many demonstrate problematic side effects. Immunotherapy may be able to overcome these limitations and further improve clinical outcomes. Similarly, emerging immunologic carcinoid biomarkers may improve diagnostic capabilities. Recent developments in immunotherapeutic and diagnostic modalities of carcinoid management are summarized here.

## 1. Introduction

The earliest written record of cancer in homo sapiens dates to 3000 B.C. [1,2]. In fact, cancer predates the evolution of homo sapiens [3]. The field of paleopathology presents evidence of neoplastic cell growth as far back as 2.4 million years ago, with the identification of osteosarcoma in the femur of the stem-turtle, Pappochelys rosinae [4]. Despite recent advances in anti-cancer therapeutics, cancer persists as a human disease, in large part because of the tumor cells’ ability to evade the human immune system [5]. By bolstering or supplementing the human immune system, emerging immunotherapeutic strategies aim to improve clinical treatment outcomes. Indeed, immunotherapies utilizing ipilimumab, nivolumab, and pembrolizumab have already demonstrated pronounced clinical utility in melanoma, head and neck squamous cell carcinoma, and cervical cancer, respectively. While immunotherapeutic interventions continue to improve clinical outcomes across a variety of cancer types, researchers are now beginning to apply immunotherapeutic strategies to the management of neuroendocrine or carcinoid tumors. Meanwhile, the incidence of neuroendocrine tumors is on the rise, with an approximate 20% increase in reported annual age-adjusted incidence between 1973 and 2004 [6].

The term “carcinoid” is conventionally understood to represent a type of neuroendocrine tumor, but its precise definition is contested. In some contexts, high-grade neuroendocrine tumors are described as neuroendocrine carcinomas, with the carcinoid term reserved for well-differentiated low-grade and intermediate-grade tumors [7]. However, these terms are frequently used interchangeably. Furthermore, terminology describing neoplasms derived from neuroendocrine cells often changes according to the anatomical location [8]. For the purposes of this review, we will employ classifications recommended by the European Neuroendocrine Tumor Society and the World Health Organization (WHO), as outlined by Kristína et al. [8]; we will also use terminology consistent with the National Cancer Institute (NCI) Physician Data Query (PDQ) database, as described in Section 2, “Carcinoid” [9]. Efforts to establish consistent terminology and tumor grading practices across multiple anatomical locations are ongoing, with the 2022 WHO Classification of Endocrine and Neuroendocrine Tumors making significant efforts to address diagnostic criteria that impact prognostication and clinical decision making; Rindi et al. recently published an excellent overview of this document, which contains helpful tables and figures summarizing these efforts [10]. In this review, we will focus primarily on adult gastrointestinal, tracheobronchial, lung, pancreatic, and ovarian carcinoid tumors to highlight emerging immunotherapeutic interventions.

The utility of the immune system as an anti-cancer tool was first conceptualized in the 19th century by Wilhelm Busch and Friederich Fehleisen, after observing spontaneous tumor regression in patients who developed the superficial skin infection erysipelas [11,12]. Later, William Coley, often referred to as the “Father of Cancer Immunotherapy”, observed a similar phenomenon when sarcoma patients with erysipelas demonstrated better outcomes than sarcoma patients without erysipelas [13]. Further advancements came in the 20th century when Lewis Thomas and Sir Frank Macfarlane Burnet independently proposed the “Cancer Immunosurveillance” hypothesis after observing the immune system recognize and target tumor-associated neoantigens to prevent carcinogenesis [11,12].

Recent advances in our understanding of the tumor microenvironment and tumor mutational burden have led to the development of clinically effective cancer immunotherapeutic agents. These agents may be classified as active or passive, depending on their ability to stimulate a sustained immune response against malignant cells. Active immunotherapeutic agents enhance the host immune response, while passive immunotherapeutic agents have intrinsic antineoplastic activity [14].

Active immunotherapies, which produce a more durable response, include checkpoint inhibitors, and oncolytic viruses. One way in which tumors have acquired mechanisms to evade our immunologic defenses is through the blockade of T-cell function and antigen recognition. This includes specific immune checkpoint pathways mediated by cytotoxic T-lymphocyte-associated antigen-4 (CTLA-4) and programmed cell death protein 1 (PD-1). While our immune system functions to protect us from threats, it is vital that it effectively differentiates self from not-self. One way in which this is accomplished is through immunologic checkpoints that prevent damage to healthy cells [15]. The process of T-cell function and antigen recognition is tightly regulated by costimulatory recognition and checkpoints processes. These checkpoints work to prevent inappropriate overactivation of the immune response. Many cancer cells evade detection by the immune system by blocking these checkpoints [16]. One such checkpoint is mediated by CTLA-4, located on the T-cell surface. CTLA-4 is a negative regulator of T-cell activation (Figure 1). CTLA-4 competes with the costimulatory molecule CD28 for shared ligands. If CTLA-4 binds to the costimulatory ligands found on both antigen-presenting cells and B cells, signal transmission is prevented, and T-cell activation is inhibited [17]. Another more recently discovered checkpoint is PD-L1 (Figure 1), which can be located on tumor cells [18].

Passive immunotherapies include monoclonal antibodies, chimeric antigen receptor (CAR) T-cell therapies (Figure 2), and antibody-dependent cell cytotoxic agents. The role of antibodies within the immune system is to neutralize a specific target. Monoclonal antibodies (mAbs) work in the same way. By binding to an antigen, mAbs block or neutralize their targets. However, due to their specificity and resistance mechanisms, mAbs are restricted in their range of activity and long-term efficacy [19]. Efforts to overcome these limitations have led to the development of bispecific antibodies (bsAbs). BsAbs are modified immunoglobulins that have two binding sites for two different antigens, or epitopes of the same antigen. This feature affords bsAbs broader application potential in the context of immunotherapy for carcinoids [20]. Additionally, bsAbs redirect immune cells to tumors, thereby decreasing the drug resistance and adverse effects seen in mAbs [21]. The role of active and passive strategies in carcinoid immunotherapy is described below.

## 2. Carcinoid

Carcinoid tumors are a rare subtype of neuroendocrine tumor that arise in a variety of different organs and soft tissue. Explicitly, neuroendocrine tumors are characterized as neoplasms that are of neuroectodermal or epithelial origin and contain neurosecretory granules. They typically show a characteristic histologic pattern and immunoprofile [21]. Though neuroendocrine tumors are considered rare, detection is becoming increasingly frequent due to improvements in screening protocols and advancements in imaging [22]. From 1973 to 2012, there was a 6.4-fold increase in the annual age-adjusted incidence of NETs in the United States [23,24].

As a subset of neuroendocrine tumors, carcinoid tumors may secrete serotonin or 5-hydroxy tryptamine (5-HT), among other bioactive peptides and neuroamines, regardless of the tumor location. Secretion of gastrin is seen in tumors involving the gastrointestinal tract [7]. Carcinoid tumors demonstrate peak incidence in the sixth and seventh decades of life, though cases in pediatric patients as young as 10 years of age have been reported [25]. While carcinoid tumors may be seen in the kidneys or genitourinary/reproductive tracts, the primary locations for carcinoids to develop involve the gastrointestinal tract (55%) and the respiratory tract (30%) or tracheobronchial tree, with the most common site being the ileum [7,26]. These neoplasms can range from well-differentiated low-grade tumors with insidious growth patterns to intermediate-grade tumors [27]. Carcinoid tumors have an incidence of 2.5–5 per 100,000 individuals per year [7]. Although carcinoid etiology is unknown, having a family history of neuroendocrine tumors or endocrine neoplasia is a risk factor [28].

Patients with carcinoid tumors may be asymptomatic or present symptoms of carcinoid syndrome, a paraneoplastic syndrome associated with carcinoid metastasis to the liver, which induces systemic release of 5-HT into the circulation; this can result in flushing, diarrhea, abdominal pain, or shortness of breath [24]. While most patients are asymptomatic, early clinical presentation of carcinoid syndrome can be insidious, with only vague reports of gastrointestinal discomfort, similar to irritable bowel syndrome or spastic colon [23]. As the disease progresses, symptoms may intensify to include flushing (seen in 90% of carcinoid syndrome patients), diarrhea (70%), or wheezing (15%), as well as myopathies and hyperpigmentation of the skin [25]. These symptoms are associated primarily with the release of serotonin, but may also be caused by histamine, kallikrein, prostaglandins E/F, and tachykinin release [24]. Duodenal carcinoid tumors may also produce gastrin to cause gastrinoma syndrome, in which there is hypersecretion of gastric acid, leading to gastroesophageal reflux disease, recurrent peptic ulcers, and chronic diarrhea [25,29]. The incidence of carcinoid metastasis is approximately 15% if the primary neoplasm is less than 1 cm in size, but can rise to 95% when tumors become greater than 2 cm in size [25].

## 3. Carcinoid Biomarkers

Because most carcinoid neoplasms are not found until they begin to elicit symptoms, detection occurs in the later stages of disease progression—after metastasis. To improve early detection and clinical outcomes, and to characterize the disease response to therapy, carcinoid biomarkers are being investigated as diagnostic and prognostic tools. Traditionally, monoanalyte biomarkers (e.g., chromogranin, serotonin, pancreastatin, etc.) were the primary carcinoid biomarkers in the management of carcinoid tumors, but the clinical utility of monoanalytes is currently obstructed by several inherent limitations, discussed below. Novel biomarkers and multianalyte tests, comprised of multiple biomarkers, are currently being developed as alternative detection and/or monitoring metrics for carcinoid tumors.

### 3.1. Carcinoid Biomarkers

A biomarker is a biological molecule contained in blood, bodily fluid, or tissue that can indicate normal biological processes or pathogenic processes. Biomarkers can be used to track disease response to treatment [30]. Serum chromogranin A (CgA) has been the most widely used biomarker in detecting carcinoid tumors [31]. Elevated levels of serum CgA have been detected in carcinoids originating from all sites of origin and exhibits diagnostic and prognostic value, despite variable background levels in different populations. Additional biomarkers used to monitor carcinoid tumors include serotonin, 5-HIAA, pancreastatin, neurokinin A, substance P, neuron-specific enolase, progastrin-releasing peptide, NT-proBNP, and connective tissue growth factor. These biomarkers correlate with the carcinoid tumor site of origin and possess specific clinical utility (Table 1). Currently, management of carcinoid tumors is limited by a paucity of accurate monoanalyte biomarkers to guide therapy response and assess disease progress. Many available monoanalyte biomarkers exhibit poor sensitivity or specificity, and do not correlate with tumor grade or differentiate low/high grade disease. The field continues to search for improved biomarkers, actively investigating new candidates, such as paraneoplastic Ma antigen 2 (PNMA2) which is detected in small intestine carcinoid tumors and used to assess the risk of recurrence [32].

### 3.2. Multianalyte Biomarkers

It is proposed that a genetic signature of an individual tumor provides superior clinical utility compared to a single biomarker [36]. Therefore, many consider the analysis of blood RNA multigene signatures to be the most advanced neuroendocrine tumor biomarker detection method [32]. Genetic signature analysis has successfully been used to identify carcinoid tumor cell activity. Analysis of 51 mRNA genetic markers via qPCR was able to distinguish gastroenteropancreatic neuroendocrine neoplasms cells from control samples [40].The neuroendocrine neoplasms test (NETest) is the first neuroendocrine tumor liquid biopsy that measures blood transcripts through qPCR to detect tumor biology and disease activity. The NETest reportedly outperforms traditional biomarkers to identify residual disease, disease progress, treatment efficacy, and accurate diagnosis [33]. Results are measured by an “activity index” ranging from 0–100. A score ≤ 20 is considered normal; values ranging from 21–40% indicate stable disease; and values between 41–100% suggest progressive disease [34,35]. The NETest, and similar genetic signature analysis methods, have the potential to detect carcinoid tumors before metastasis, thereby improving disease prognosis and treatment outcome.

### 3.3. Carcinoid Immune Biomarkers

The type and degree of immune cell infiltration varies, depending on the site of the primary carcinoid tumor. Analysis of the immune landscape and tumor mutational burden, specifically programmed death-1 (PD-1) and programmed death-ligand 1 (PD-L1) protein expression, may be useful to assess immunotherapy treatment response related to carcinoid.

#### 3.3.1. Gastrointestinal Carcinoids

A study of 68 patients diagnosed with midgut carcinoid tumors reported an elevated circulating regulatory T-cell count and an associated decrease in T-cell proliferative capacity. Tumors were infiltrated by CD4+ and CD8+ T cells in the presence of CD4+ FOXP3+ cells; heavier infiltration was detected in metastatic compared to primary midgut tumors. Both systemic Th1-promoting cytokines and proinflammatory cytokines IL-20p70 and IL-1B were significantly reduced, whereas IL-8 was significantly elevated in participants [41]. The results of this study suggest that the immune system can detect that the presence of carcinoid cells is abnormal and attempts to mount an immune response, given the increase in inflammatory cytokines and regulatory T cells. However, the immune response is blunted, as tumor cells are not spontaneously killed by the infiltrating T cells.

In another study, the immune microenvironments of 102 resected, small bowel, carcinoid tumors were analyzed for expression of PD-L1. The study detected mutation in 40 of the 102 tumors. PD-L1 is normally expressed by tissue to suppress the inflammatory response and limit tissue damage [42]. When expressed by tumor cells, PD-L1 will bind PD-1 on the surface of activated T cells, leading to T-cell neutralization and cancer cell survival [43]. Compared to jejunal- and ileal-originating carcinoid tumors, PD-L1 clone 28-8 expression was significantly elevated in duodenal carcinoid tumors. Approximately 66% of tumors had inflammatory cell infiltration, and intratumor infiltration was positively associated with PD-L1 expression. This suggests tumor-infiltrating lymphocytes release cytokines that upregulate PD-L1 expression as an adaptive immune-resistance mechanism. There was no correlation of PD-L1 expression or the degree of immune infiltration with overall survival [44]. The infiltration of carcinoid tumor cells by lymphocytes may increase PD-L1 expression to neutralize T cells and promote cancer cell survival. Suppression or blockage of PD-L1 expression by tumor cells could lead to a significant immune response and destruction of carcinoid tumor cells.

#### 3.3.2. Pulmonary Carcinoids

The composition and frequency of tumor-infiltrating leukocytes using flow cytometry of resected, low-grade, pulmonary, typical carcinoid tumors from four never-smoker females revealed a variety of infiltrating immune cells. By abundance, 19.8% CD8+ T cells, 17.7% CD4+ T cells, 11.5% B cells, 11% macrophages, 8.8% NK cells, 3.9% neutrophils, 1.4% dendritic cells, 1.2% basophils, 0.8% mast cells, and 0.6% eosinophils were detected. It was determined that a complex immune landscape existed that was relatively non-inflammatory [45]. PD-1 clone SP269 and PD-L1 clone E1L3N expression from 131 typical and 37 atypical pulmonary carcinoid tumors was also investigated. High PD-1 expression was detected in 16% of all tumors, PD-L1 was detected in 7% of typical carcinoid tumors, and all atypical carcinoid tumors were PD-L1-negative. PD-L1 was associated with metastatic disease [46]. The presence of tumor-infiltrating leukocytes and immunosuppressive mutations in carcinoid neoplasms suggests that the incorporation of immune-modulating therapies with current treatment options may improve clinical outcomes by directing an immune response to the tumor.

## 4. Non-Immune Therapy for Carcinoid

The selection of treatment for carcinoid is determined based on a number of factors, including the primary location, rate of growth, and tumor expression of particular receptors [7]. Current non-immune therapies mainly focus on partial to full resection of tumor through surgery and pharmacological treatment.

### 4.1. Advantages and Disadvantages of Surgical Intervention

Surgical resection can be curative and is the gold standard of treatment for resectable carcinoids of the small intestine and lung. However, the success of surgical intervention heavily depends on the physical characteristics of the tumor, such as its size and location [47]. Metastasis of the tumor, as seen with lymph node involvement, is a major indicator of poor prognosis. In a study of 106 patients with surgically resected carcinoids of the lung, 24% experienced metastasis and a 25% mortality after 6.5 years, with metastasis being the strongest negative prognostic factor. Within the same cohort of lung carcinoid patients, 5% experienced local tumor relapse [48]. This emphasizes that although surgery may be a viable treatment option for tumors localized to a specific region, surgery alone is insufficient in the treatment of carcinoid tumors with systemic metastasis [47]. Other factors to consider when selecting surgical treatment for carcinoids are procedure-specific complications, such as infection, bleeding, and death [47]. For many patients, recovery time can also increase overall morbidity, especially given the older age of many carcinoid tumor patients [47]. Though these risks have decreased with the advancement of minimally invasive procedures, the issue of tumor recurrence is still apparent. While surgery may be beneficial for isolated tumors or as part of a combined approach, surgery monotherapy may be an incomplete tactic. Pharmacotherapeutic options offer some advantages over surgical intervention, but pharmacotherapy also has relative strengths and weaknesses.

### 4.2. Advantages and Disadvantages of Pharmacotherapy

Randomized trials have investigated the use of chemotherapeutic agents to treat carcinoid tumors (Table 2). A trial evaluating capecitabine paired with temozolomide (CAPTEM) in a cohort of 116 carcinoid patients with pancreatic-, lung-, and small bowel-origin NETs demonstrated an overall response rate of 21%, with adverse events present in 64% of cases. These side effects included nausea and fatigue as well as thrombocytopenia [49]. Other immunotherapeutic options are summarized in Table 3. Everolimus is a direct inhibitor of the PI3K/AKT/mTOR signaling pathway that plays a key role in the pathogenesis of the NETs [50]. In the setting of NETs, mTOR inhibitors demonstrate a decrease in growth factor signaling, metabolism, and variations within the genome, which activate the mTOR pathway. This leads to a delay in NET proliferation [51]. Like both the somatostatin analogue (SSA) and chemotherapeutic agents, everolimus has demonstrated antitumor activity, and even higher efficacy in more aggressive tumors. However, it is still limited in the number of sites it can effectively treat. The main pharmaceutical treatment in well-differentiated gastroenteropancreatic NETs is SSA therapy, particularly in somatostatin-receptor-(SST)-positive tumors. SSAs serve as the first-line therapy for the treatment of carcinoid syndrome caused by midgut carcinoids due to the high expression of SST. In addition to treating tumor-associated symptoms, such as refractory diarrhea, SSA therapy also has an antiproliferative mechanism of action [52]. However, these agents have been associated with biliary stasis and cholelithiasis, and additional medical interventions, such as a cholecystectomy, may need to be performed. Somatostatin analogues are also known to cause pancreatic malabsorption, which can necessitate pancreatic enzyme supplementation [7].

In many cases, when treating carcinoid tumors, a multi-regimen approach is necessary. For instance, surgical removal of a tumor may still require somatostatin analogues to treat the flushing and diarrhea associated with malignant carcinoid syndrome [53]. Though SSAs may mitigate metastasis, they are also limited in their scope of treatment; SSAs have only been reported to address metastasis of well-differentiated tumors [52]. This finding, alongside the fact that SSAs are very specific in their targeting, emphasize their restrictions as a broad-spectrum treatment against carcinoid tumors. The interest in immunotherapeutic approaches continues to increase, as immunotherapeutic agents may have the potential to overcome shortcomings associated with traditional non-immune therapies [54].

**Table 2 molecules-28-02047-t002:** Randomized trials of chemotherapeutics for Carcinoid Tumors.

Intervention	Results
STZ with cyclophosphamide [55]	Carcinoids primary to small bowel: overall response rate (ORR) 37% Carcinoids of pulmonary or unknown region: ORR 17%
STZ with 5-flourouracil (5-FU) [56]	Metastatic carcinoid tumors: ORR 22%
STZ with doxorubicin [57]	Advanced carcinoid tumors: ORR 16%
Recombinant IFN-alpha-2a [58]	Metastatic carcinoid tumors: Progression-free survival median of 14.1 months
Capecitabine paired with temozolomide [49]	Pancreatic, lung, and small bowel-origin NETs: ORR 21%

**Table 3 molecules-28-02047-t003:** Immunotherapeutics for Carcinoid Tumors.

Immunotherapy	Target	Carcinoid Typed
Combined Ipilimumab/Nivolumab	CTLA-4/PD-1	Locations 32 patients: 18 with high-grade disease, 10 with intermediate-grade disease, and 4 with low-grade disease. Gastrointestinal (GI): 15, Lung: 6 NCT02834013 [59,60]
Pembrolizumab	PD-1	25 PD-1-positive advanced or metastatic carcinoid tumors. Lung: 9, GI: 7, Other: 9 NCT02054806 [61]
Pembrolizumab	PD-1	GI tumors: 14, Pancreatic NETs: 8 NCT03043664 [62]
Peptide Receptor Radionuclide Therapy (PRRT) 177 Lu-Dotatate	SSTR (somatostatin receptor)	Midgut carcinoid tumors NCT01578239 [63,64]
Spartalizumab (PDR001)	PD-1	Advanced NETs from pancreatic, GI, and thoracic origins including 116 pts with well-differentiated NETs NCT02955069 [65]
Tidutamab (previously XmAb18087)	SSTR2 and CD3	Advanced NETs including 41 participants comprised of the following: 46% pancreas, 22% intestine, 20% lung, and 12% GEP-NET/unknown NCT03411915 [66]
AdVince	Recombinant Adenovirus	Treat liver metastases from NETs including metastatic midgut carcinoids that express Chromogranin A NCT02749331 [67]

## 5. Immunotherapy for Carcinoid

### 5.1. Overview of Immunotherapy and Carcinoid [27]

The rising interest in immunotherapy as a means of cancer therapy is rooted in its precision compared to the nonspecific approaches seen with surgery and chemotherapy. Active immunotherapy involves the direct stimulation of the immune system against the tumor, resulting in immunologic memory. It has been reported to have benefits in melanoma and lung cancer, as well as Hodgkin’s lymphoma [68]. Infusion of passive immunotherapies provide the host immune system with antibodies and cytokines that target cancer cells and elicit an immune response with limited duration. This form of therapy has previously demonstrated beneficial effects in metastatic melanoma [68]. Due to both therapies working within the immune system, they both have a wide range of cancers in which they can be utilized. Additionally, the restoration of the immune response seen with immunotherapy is able to increase the body’s ability to fight against tumor recurrence and metastasis, which numerous traditional therapies have difficulty protecting against [69]. On the other hand, there have been limitations in the use of immune therapies, primarily due to slow growth, low mutational burden, and the failure of carcinoid tumors to provoke a strong immune response [27].

### 5.2. Active Immunotherapy for Carcinoid

Active immunotherapy is the administration of specific agents to elicit a patient’s anti-tumor immune response [68]. These therapeutic agents include immune checkpoint inhibitors and oncolytic vaccines, which decrease the tumor cell’s ability to evade the cytotoxic effects of T cells [68]. This mechanism has shown to be beneficial in anti-tumor therapy, with efficacy in melanoma, lung, kidney and bladder cancers, while also increasing the proliferation of tumor-infiltrating lymphocytes. However, there are associated drawbacks that impose limitations on long-term treatment. In addition to high cost and unacceptable toxicities that may occur through activation of the immune system (also seen in passive immunotherapy), there have been issues regarding resistance producing negative regulation, resulting in autoimmune diseases and death [69].

#### 5.2.1. Ipilimumab/Nivolumab

Ipilimumab, a humanized monoclonal anti-CTLA-4 antibody, was the first-in-class immune checkpoint inhibitor approved for the treatment of cancer [16]. In a prospective, open-label, multicenter phase II clinical trial (NCT02834013), the efficacy of ipilimumab plus nivolumab, a PD-1 inhibitor, is being evaluated in atypical bronchial carcinoid tumors, neuroendocrine carcinoma, and grade 3 NETs independent of the primary site [59]. In total, 32 eligible patients received the therapy, and of that, 18 patients had high-grade disease. The most common sites were gastrointestinal (*n* = 15) and lung (*n* = 6). The overall response rate (ORR) was 25% (CR 3% 1 pt, PR 22% 7 pts); those with neuroendocrine carcinoma demonstrated an ORR of 44% in patients with nonpancreatic high-grade neuroendocrine carcinoma, with 0% ORR in low/intermediate grade disease [59]. Seeking to validate these findings within the high-grade neuroendocrine neoplasm cohort, a second prospective study reporting ipilimumab plus nivolumab demonstrated a 26% ORR in patients with high-grade neuroendocrine neoplasms, with durable responses seen in patients with refractory disease [60]. The response rates can be compared to rates seen in randomized trials involving chemotherapeutic treatment of carcinoid tumors in Table 2. The combination of ipilimumab and nivolumab had limited to no efficacy for the treatment of carcinoid tumors. Current data suggest that the combination immunotherapy has a more prominent treatment role in metastatic neuroendocrine carcinomas.

#### 5.2.2. Pembrolizumab

Pembrolizumab (Keytruda) is a humanized monoclonal anti-PD-1 antibody that has been broadly studied across a wide variety of neoplasms [70]. The Pembrolizumab with Lanreotide Depot for Gastroeneteropancreatic Neuroendocrine Tumors (PLANET) clinical trial (NCT03043664) investigated the effects of pembrolizumab with lanreotide, a somatostatin analogue with anti-tumor and serotonin suppression effects in patients with nonresectable, recurrent, or metastatic well-/moderately differentiated gastroenteropancreatic neuroendocrine tumors (GEP-NETs). In total, 22 patients were treated—14 with gastrointestinal tumors and 8 with pancreatic tumors. Of the 12 GEP-NETs analyzed thus far, 4 contained datable PD-L1 expression. In this population, the combination of pembrolizumab and lanreotide achieved stable disease in approximately 40% of patients [62].

The KEYNOTE-028 study (NCT02054806) examined the efficacy and safety of pembrolizumab in biomarker-positive solid tumors, including 170 patients with PD-L1-positive advanced or metastatic carcinoid tumors, in addition to 106 patients with well- or moderately differentiated pancreatic NETs (pNETs). Of that, 21% of the well-differentiated pNETs and 25% of the moderately differentiated pNETs were PD-L1-positive tumors, being PD-L1+ carcinoid tumors, and 16 being PD-L1+ pNETs receiving treatment. For PD-L1 carcinoids receiving treatment, the ORR was 12% and 6.3% in the pNET group. Overall, this study shows that pembrolizumab demonstrates anti-tumor activity in a subset of NET patients [61].

The KEYNOTE-158 study (NCT02628067) investigated pembrolizumab in a larger cohort of patients, 1595 total participants, with advanced solid tumors. Included in this study is a total of 107 NET patients involving the lung, appendix, small intestine, colon, rectum, or pancreas who received treatment, 15.9% of whom had PD-L1-positive tumors. The ORR was 3.7%, demonstrating limited antitumor activity with pembrolizumab monotherapy [67]. The PLANET and KEYNOTE studies concluded that pembrolizumab treatment can achieve stable disease in patients with carcinoid tumors, but has a poor ORR compared to the chemotherapeutic agents reported on in Table 3.

#### 5.2.3. Spartalizumab

Spartalizumab is a humanized monoclonal anti-PD1 antibody whose efficacy was studied in patients with well-differentiated metastatic NETs and poorly differentiated gastroenteropancreatic neuroendocrine carcinomas (GEP-NECs). The study (NCT02955069) included 95 patients with well-differentiated NETS and 21 in the GEP-NEC groups; the ORR was 7.4% in the NET group and 4.8% in the GEP-NEC group. Both groups fell below the predefined success criteria of ≥10%. However, it is worth mentioning that in those with thoracic NETs (*n* = 30), the ORR was 16.7%. Researchers concluded that the efficacy of spartalizumab is limited in this heterogenous group, although its efficacy with thoracic NETs warrants further investigation [71].

### 5.3. Passive Immunotherapy for Carcinoid

Passive immunotherapy is often referred to as adoptive cell transfer (ACT) or cell-based therapy. Rather than stimulating a patient’s immune system directly, as seen in active immunotherapy, passive immunotherapy focuses on extracting a patient’s lymphocytes (primarily T lymphocytes and natural killer cells) and altering them ex vivo, such that they become capable of attacking specific neoantigens. The cells are then reintroduced into the patient’s circulation, with the release of cytokines and cytolytic actions to destroy specific tumor cells [5]. Passive immunotherapy has been shown to have success in direct lymphocytic killing towards NETs based on various markers, such as SSTR2+ receptor, as well as multiple lymphocytic leukemias and lymphomas [5]. However, this form of therapy is more cumbersome in production compared to active immunotherapy due to its extraction–reinfusion process, which also decreases its reproducibility in targets, as well as in other patients.

#### 5.3.1. Tidutamab

Tidutamab (Figure 3), previously known as XmAb18087, is a monoclonal bispecific antibody for SSTR2, a somatostatin receptor, and CD3 [66]. CD3 is an intracellular signaling domain in T cells that relays antigen engagement information from T-cell binding T-cell receptors [72]. The combination of the anti-SSTR2 and anti-CD3 activity of Tidutamab allows T-cell-mediated cytotoxicity directly towards SSTR2+ cells, often overexpressed in many NETs. In a phase I, multiple-dose, ascending-dose, escalation clinical trial (NCT03411915), the recommended dose, safety, efficacy, and anti-tumor activity of Tidutamab is being analyzed in 87 participants (42 at the time of this writing) with advanced NETs and gastrointestinal stromal tumors (GIST). Preliminary data as of August 2021 shows that Tidutamab is well-tolerated in solid tumors, with low incidence (41%) of cytokine-release syndrome, all recovered. There is a stable disease achieved in 26.8% of patients; however, a high PD-L1 clone E1L3N expression, especially seen in peripheral T cells, associates with poor treatment outcomes [66]. The association of high PD-L1 expression and poor treatment outcomes was unexpected. As mentioned earlier, monoanalyte biomarkers, such as PD-L1, can be inaccurate when predicting the response to immunotherapy. There are a multitude of PD-L1 clones, and their variation may lack comparability when assessing treatment outcome. Although Tidutamab can facilitate the recruitment of T cells and their binding to carcinoid tumor cells, the correlation of PD-L1 expression and poor treatment outcomes suggests that T cells are inactivated by PD-L1-positive tumor cells. The combination of Tidutamab and PD-1 or PD-L1 antibodies could result in a better treatment response.

#### 5.3.2. ^177^Lu-Dotatate

Peptide receptor radionuclide therapy (PRRT) targets tumors using radioactive particles that bind specific receptors to deliver a localized dose of radiation. ^177^Lu-Dotatate is an example of PRRT, where a patient receives an injection of Octreotide/Octrotate (Dotatate) that is radiolabeled with lutetium-177. This therapy has been linked to treating NETs expressing somatostatin receptors. A phase III, multicenter, open-label clinical trial (NCT01578239), consisting of 229 patients with well-differentiated metastatic midgut NETs, is testing the safety and efficacy of ^177^Lu-Dotatate [63]. Primary analysis shows that after month 20, there is a 65.2% progression-free survival rate. Additionally, less than 10% of cases have reported clinically significant myelosuppression. The study demonstrated that ^177^Lu-Dotatate has great potential in treating metastatic GI NETs, while minimizing side effects typically seen with carcinoid therapies.

## 6. Conclusions

Emerging immunotherapeutic options that may be effective for the treatment of carcinoid tumors encompass both active and passive immunotherapeutic modalities. Ongoing efforts demonstrate the potential for immunotherapeutic treatment of carcinoid tumors. Anti-PD-1 antibody pembrolizumab paired with somatostatin analogue lanreotide produced stable disease in approximately 40% of patients. Tidutamab, by way of antibody-dependent cell toxicity, achieved stable disease in 26.8% of patients. Positive short-term responses to PRRT ^177^Lu-Dotatate include a progression-free survival rate of 65.2% after month 20. Several immunotherapies did not achieve the targeted response rates in their respective studies. Ipilimumab, a humanized monoclonal anti-CTLA-4 antibody, plus nivolumab, a PD-1 inhibitor, demonstrated an ORR of 0% in low- and intermediate-grade carcinoid tumors. Spartalizumab’s efficacy was limited in the heterogenous group of carcinoid tumors, but its efficacy for the treatment of thoracic NETs warrants further investigation. Furthermore, higher ORRs were achieved by immunotherapies that acted against high-grade neuroendocrine carcinomas. Ipilimumab plus nivolumab demonstrated an ORR of 44% in patients with nonpancreatic high-grade neuroendocrine carcinoma. Overall, these findings suggest that immunotherapies are more effective against aggressive neuroendocrine tumors with rapid growth rates not characteristic of carcinoid tumors. The long-term response rates and scope of adverse events associated with immunotherapeutic agents are still uncertain and require further evaluation. Immunotherapy for the treatment of carcinoid tumors may still prove to be a useful treatment modality, particularly in combination with surgery and other pharmacologic regimens, such as somatostatin analogues. Future focus on alternative immunotherapeutic targets, such as macrophage targeting therapy, cytokine therapy, TLR agonists, checkpoint inhibitors, and T-cell-targeting nanoparticles, may provide productive avenues of anti-carcinoid immunotherapeutic advancement.

## 7. Methods

PubMed database queried for review articles and randomized controlled trials in the English language spanning 1978 to 2023. The following terms were used to generate searches: carcinoid tumors; immunotherapy in carcinoid tumors; neuroendocrine tumors; carcinoid tumor nomenclature; carcinoid treatments; carcinoid chemotherapy; carcinoid biomarkers; carcinoid surgical interventions.

## Figures and Tables

**Figure 1 molecules-28-02047-f001:**
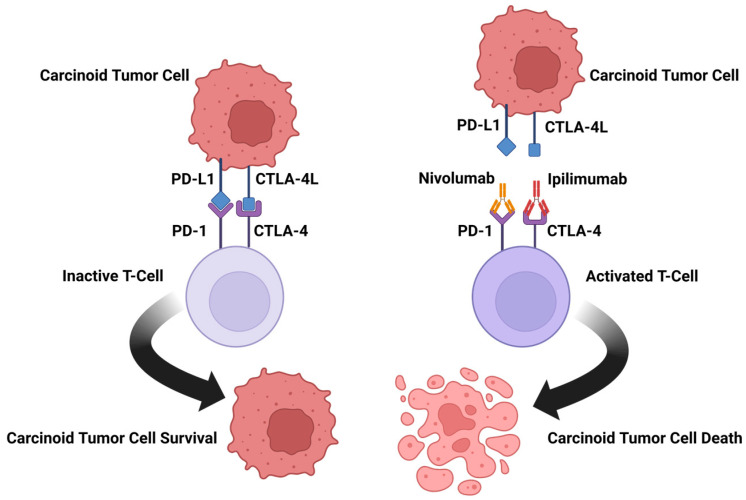
Mechanism of action for the combination of ipilimumab and nivolumab. Carcinoid tumor cells can express PD-L1 and CTLA-4L. These ligands can bind PD-1 and CTLA-4 to inactivate T cells. By blocking the PD-1 and CTLA-4 receptors on T cells, the T cells remain active and capable of mounting an immune response against carcinoid tumor cells. In this way, the combination of ipilimumab and nivolumab suppress this type of cancer immune escape.

**Figure 2 molecules-28-02047-f002:**
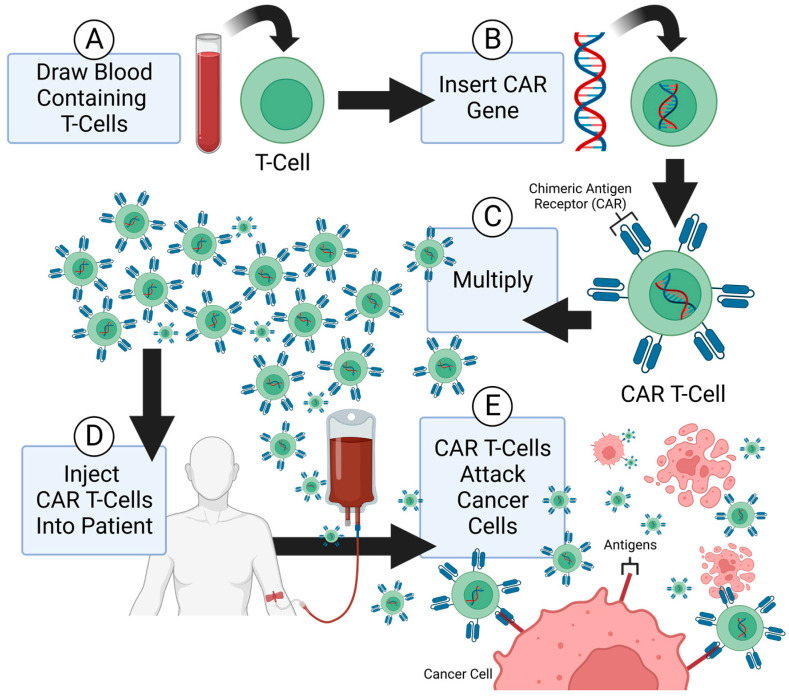
Overview of chimeric antigen receptor (CAR) T-cell therapy. (**A**) Initial steps include obtainment of T cells from a blood sample, followed by (**B**) insertion of CAR genes into T cells with subsequent expression of CAR on the T-cell membrane surface. Next, (**C**) T-cell multiplication is induced, followed by (**D**) insertion of the CAR T cells into the patient. (**E**) CAR T cells then bind to specific antigens located on the membrane surface of cancer cells and induce an immune response.

**Figure 3 molecules-28-02047-f003:**
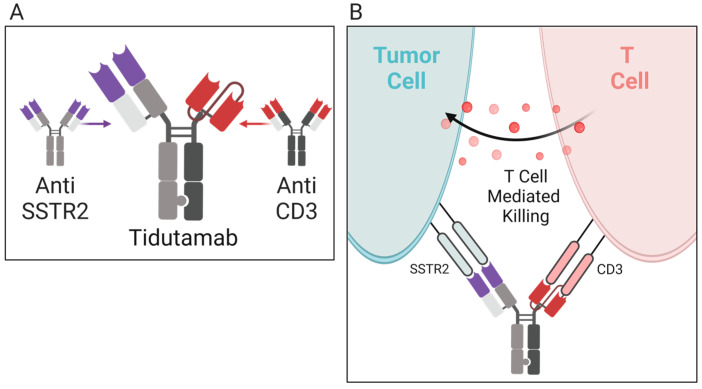
Mechanism of action for tidutamab. (**A**) Tidutamab is a bispecific antibody comprised of anti-somatostatin receptor (anti-SSTR2) and anti-CD3 found on T cells. (**B**) When activated, SSTR and CD3 trigger T-cell-mediated cytotoxicity. The tidutamab antibody combination facilities T-cell-mediated cytotoxicity against carcinoid tumor cells expressing the SSTR2 receptor.

**Table 1 molecules-28-02047-t001:** Monoanalyte Carcinoid Biomarkers.

Biomarker	Carcinoid Location	Assess/Correlations	Sensitivity and Specificity
Chromogranin A(CgA)	All locations	Confirm diagnosis, assess treatment progress, tumor burden, correlated to tumor load, background levels variable in different populations [32,33,34,35]	43–100% sensitivity 10–96% specificity [36]
Serotonin (5-HT)	Foregut, Midgut	Blood serum analysis, carcinoid syndrome [32,37]	35% sensitivity up to ≈100% specificity [36]
5-HIAA	Midgut	Urinary or serum analysis, carcinoid syndrome, used for screening and diagnosis [32,36]	35% sensitivity Up to ≈100% specificity [36]
Pancreastatin	Pancreas, Midgut	Tumor activity [32,36,38]	64% sensitivity 58–100% specificity [36]
Neurokinin A (NKA), Substance P	Midgut	Prognostic value, correlated with poor outcome [36]	88% sensitivity No data for specificity [36]
Neuron-specific enolase (NSE)	All locations	Elevated levels suggest poor differentiation [32,36]	33% sensitivity Up to 100% specificity [36]
Progastrin- releasing peptide (proGRP)	Lung	Expression associated with survival, >90 ng/L negatively correlated with outcome [36]	99% sensitivity 43% specificity [36]
Pancreatic Polypeptide (PP)	Pancreas, Midgut, Colon	No known clinical utility [36]	50–80% sensitivity No data for specificity [36]
N-terminal pro-brain natriuretic peptide (NT-proBNP)	Midgut	Prognostic value, correlates with survival in carcinoid heart disease [36]	87% sensitivity 80% specificity [36]
Connective Tissue Growth Factor (CTGF)	Midgut	Elevations predict reduced right ventricular function in carcinoid heart disease [36]	88% sensitivity 69% specificity [36]
Paraneoplastic Ma antigen 2 (PNMA2)	Small intestines, Lung	Assess recurrence risk [39]	46–50% sensitivity SI-NETs 35% sensitivity lung 98% overall specificity [39]

## Data Availability

The authors confirm that the data supporting the findings of this study are available within the article.

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
