# Peer review of "Emerging Immunotherapeutic and Diagnostic Modalities in Carcinoid Tumors"

_molecules, 2023, doi:10.3390/molecules28052047_

Round 1

Reviewer 1 Report

Review report

Article number : molecules-2027884

Title : Emerging Diagnostic and Therapeutic Modalities in Carcinoid Tumors

Abstract

Line 9 and 10 : “Cancer evasion of our immune system is an oncogenic strategy employed by tumors”, rewrite as “ Envasion of innate immunity is the strategy employed by tumor cells “

Line 11 : “applied to” – replace with “utilized for”

Line 13-14:  “Though surgical intervention can be curative, tumor characteristics such as size, location, and spread heavily limit the success surgical resection can achieve”, Please restructure the sentence as idea conveyed in ambiguous.

Line 17: Delete “The” at the beginning

Introduction

Line 22 : “Cancer precedes the human immune system. In fact, cancer precedes humankind” : Avoid such incomplete and fragmented sentences

Line 23: “The field of paleopathology presents evidence of neoplastic cell growth as far back as 2.4 million years ago” :      in which species??

Line 25-27 : “ Despite recent advances in molecular biology and genomics, cancer persists as a human disease, in large part because of its ability to evade immune clearance. Even when major therapeutic intervention results in disease remission, disease recurrence persists” : Restructure the sentence.

Line 30: “such as” : replace with “utilizing”

Line 34: 19th : use superscript

Line 35 : Replace “that” with who

Lot of corrections as above; require language editing.

Line 88: “For the purposes of this review, we will employ classifications recommended by the European Neuroendocrine Tumor Society and the World Health Organization, as outlined by Kristína et. al [17]; we will also use terminology consistent with the National Cancer Institute(NCI) Physician Data Query (PDQ) database”: how they define and differentiate carcinoma from carcinoid?

Carcinoid

Line 118- “carcinoid tumors may secrete serotonin and gastrin, among other bioactive peptides and neuroamines” : From every NET? Or in stomach only? Specify.

At many places it is used as “serotonin” . Though it is technically correct, is more appropriately known now a days as “5-HT” or 5-hydroxy tryptamine.

Line 141 : Define what is carcinoid syndrome

Line 152-158 : distinguish between monoanalyte and multianalyte biomarkers with examples

Line 180: explain “RNA multigene signatures” with examples like “70 gene signature” or similar.

Line 182: QT-PCR : Use either as qPCR or qRT-PCR

Table 1:

1. Use a proper self-explanatory table title

2. Include the specificity and sensitivity of biomarkers mentioned in the table

Section 3.3.1 Gastrointestinal Carcinoids : This section only reporting two papers, without any conclusion. Inference by the authors, as to what exactly is indicated by this study is missing

Section 3.3.2 : Pulmonary Carcinoids- Same as above. A section in the review article is supposed to covey a particular conclusion by authors, which is to be substantiated with the already published research works. All the papers that are referred should be consistent and highlight a concept, rather than mere review of literature with some discrete publications

Section 4. Non-Immune Therapy for Carcinoid: This section may be a little bit elaborated, with sub headings on advantages and disadvantages of surgical and pharmacotherapy of carcinoids

Table 2 and 3: Renumber the tables according to their occurrence in the body text. Pharmacotherpy is occurring second, but the corresponding table number is 3, Though immunotherapy is third, the table number is 2

Section: 5.2 Active Immune Therapy for Carcinoid : Change to immunotherapy

Section 5.2 Active Immune Therapy for Carcinoid : Before starting abruptly with the agents, it is better to have some introductory paragraph, mentioning what is immunotherapy, its advantages and disadvantages and commonly employed agents in the treatment of carcinoids. Currently adopted way of writing misses the continuity of section from previous sections

Line 332: 22 patients were treated in total—14 with gastrointestinal tumors and 8 with pancreatic tumors. Of the 12 tumors analyzed, 4 contained datable PD-L1 expression:  Are the 12 from GI Pancreatic or both? If both, how many from each? Specify reading materials and methods

Line 324: In this population. the combination of pembrolizumab and lanreotide produced stable disease in approximately 40% of patients [66] : Is the combination used for producing the disease??

As mentioned previously, the conclusion by the authors is missing. For example, after reading section 5.2.2 Pembrolizumab, it is not clear whether the treatment with Pembrolizumab has some advantage over surgical or pharmacological interventions, considering that the purpose of this review is to prove the advantages of immunotherapy over the other two.

Section 5.2.4 Oncolytic viruses: This is another big class of Non pharmacological control measure (But no application in diagnosis, which is one of the theme of this review). Either it is to be elaborated as a separate standalone section, or it is to be deleted.

Section 5.3 Passive immune therapy for carcinoid: Before starting abruptly with the agents, it is better to have some introductory paragraph, mentioning what is passive immunotherapy, its advantages and disadvantages over active immunotherapy and commonly employed agents in the treatment of carcinoids. Currently adopted way of writing misses the continuity of section from previous sections

What exactly is the difference between active and passive immunotherapy is not mentioned.

The major corrections required while rewriting the paper

1. The title of the review is “Emerging Diagnostic and Therapeutic Modalities in Carcinoid Tumors” but the diagnostic part is poorly discussed.

2. As per the abstract, the authors try to prove the advantages of immunotherapy for carcinoids. But in the section 5.2. and 5.3. they fail to substantiate the same.

3. None of the sections have any contribution by the authors. In a review article, the observations and conclusions of the authors, which are to be substantiated by published works is highly essential, rather than being a collection of randomly selected publications.

4. At many places, the flow of ideas is lost. Many sub sections start abruptly, without a proper introduction to the topics of discussion under that title.

5. English language editing is mandatory. Many sentences are ambiguous or confusing

6. Titles of table and figures are to be self-explanatory, which is not so in almost all tables and figures

7.At many places, the important terms are not properly defined at its first occurrence.

8. Some sentences especially in the initial part of the article are fragmented and with improper or no required punctuations

Reviewer 2 Report

The strengths of the review are its broad coverage of immunotherapeutic options and biomarkers for carcinoid tumors including the NETest. However, I think the manuscript would significantly benefit from re-organization and revision of the text.

1.If the authors are choosing to focus on immunotherapeutic interventions in their review, the title should reflect that.

2.The introduction primarily focuses on the background of immunotherapy with the first mention of carcinoid tumors on page 2 line 73. It should be rewritten to emphasize the importance of carcinoid tumor therapeutics based on outcomes data and the current gaps in diagnostics and therapeutics for carcinoid tumors.

3.Although the authors describe terminology issues surrounding neuroendocrine tumors there should be an attempt to clarify them with a table or figure. There should be mention of the current 5th ed. of the WHO Classification of Neuroendocrine Tumours and the move toward unified terms and grading of neuroendocrine tumors across different organ sites. While carcinoid and atypical carcinoid are considered acceptable terms, diagnosis and grading should be discussed as it relates to prognosis and clinical behavior.

4.The manuscript should address the current state of knowledge regarding the pathogenesis and biology of carcinoid tumors. What is known about the molecular landscape of carcinoid tumors? What is the rationale behind targeting the PI3K/AKT/MTOR pathway?

5.What are the complex pathways associated with carcinoids that is referred to on page 2 line 73?

6.It is stated that carcinoid tumors are “immunologically cold” on page 8 line 287. Please explain.

7.The figures should be numbered in the order that they are introduced in the text.

8.Figure 2 should referenced in the text after carcinoid tumors are described since the tumor cells are specifically labeled as carcinoid tumor cells in the figure.

9.PD-1/PDL-1 expression is notoriously inconsistent in predicting response to immunotherapy. It should be noted that these are imperfect biomarkers with different antibody clones that may not be comparable in all situations. For the clinical trials mentioned that use PD-1 or PDL-1 as a biomarker, the specific antibody clone should be stated. What can be said about the poor treatment outcomes with tidutamab in patients with high PDL-1 (page 10 line 379)?

10.Where is table 3 that is referenced on page 9 line 313?

11.The conclusions seem cursory. The ORR of ipilimumab plus nivolumab in high-grade neuroendocrine carcinomas is mentioned but those are not carcinoid tumors.
